# KEBAP: Korean Error Explainable Benchmark Dataset for ASR and Post-processing

**Seonmin Koo**[1*], **Chanjun Park**[2*], **Jinsung Kim**[1], **Jaehyung Seo**[1]
**Sugyeong Eo**[1], **Hyeonseok Moon**[1], **Heuiseok Lim**[1†]
[1]Korea University, Department of Computer Science and Engineering
[2]Upstage AI
{fhdahd, jin62304, seojae777, djtnrud, glee889, limhseok}@korea.ac.kr
chanjun.park@upstage.ai

## Abstract

Automatic Speech Recognition (ASR) systems are instrumental across various applications, with their performance being critically tied to user satisfaction. Conventional evaluation metrics for ASR systems produce a singular aggregate score, which is insufficient for understanding specific system vulnerabilities. Therefore, we aim to address the limitations of the previous ASR evaluation methods by introducing the Korean Error Explainable Benchmark Dataset for ASR and Post-processing (KEBAP). KEBAP enables comprehensive analysis of ASR systems at both speech- and text levels, thereby facilitating a more balanced assessment encompassing speech recognition accuracy and user readability. KEBAP provides 37 newly defined speech-level resources incorporating diverse noise environments and speaker characteristics categories, also presenting 13 distinct text-level error types. This paper demonstrates detailed statistical analyses of colloquial noise categories and textual error types. Furthermore, we conduct extensive validation and analysis on commercially deployed ASR systems, providing valuable insights into their performance. As a more fine-grained and real-world-centric evaluation method, KEBAP contributes to identifying and mitigating potential weaknesses in ASR systems.

## 1 Introduction

Automatic speech recognition (ASR) is a task that recognizes speech and converts it into text, and it is getting more and more attention with the development of voice interface applications and devices such as Alexa, Siri, and Cortana (Williams and Young, 2007; Wang et al., 2018, 2020). In the real world, the ASR result has a trade-off between recognition accuracy[1] and user readability. Even

---

∗ Equally contributed, ‡ Corresponding author
[1]Recognition accuracy is the measure of accurately perceiving phonemes as they are externally expressed, regardless of user input quality (Liao et al., 2022).

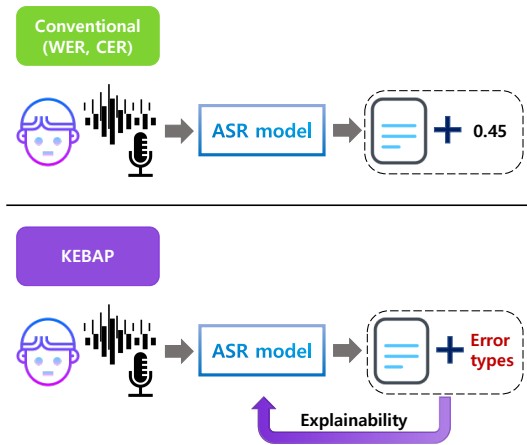

Figure 1: An example of using **KEBAP** to evaluate weaknesses in ASR results. The above approach with conventional evaluation methods lacks explanatory power. Evaluation using **KEBAP** below enables a detailed description of the weaknesses.

if the ASR model accurately recognizes the input voice, the user's readability may decrease. This is because humans do not always utter perfect sentences in the real world (e.g., incomplete utterances, sighs, etc.). To achieve balanced ASR results in this trade-off situation, it is required to consider both recognition accuracy and user readability.

In terms of recognition accuracy, various ASR evaluation metrics such as word error rate (WER) (Woodard and Nelson) and character error rate (CER) (Morris et al., 2004) have been prevalent. Also, readability is considered in ASR post-processing (ASRP) tasks, where the goal is to improve the clarity and comprehension of speech recognition outputs without modifying the underlying model architecture (Mani et al., 2020b; Liao et al., 2020; Leng et al., 2021). The ASRP task relies on quantitative metrics, such as BLEU (Papineni et al., 2002) and GLEU (Napoles et al., 2015), similar to the ASR evaluation using WER and CER.

However, it is crucial to recognize that even if

quantitative evaluation scores are similar, the qualitative aspects of the ASR results may not necessarily align. Conventional research methods, which focus on accuracy or user readability, compute quantitative scores based on the degree of alignment between inputs and outputs. This approach falls short in classifying potential error types or pinpointing the model's specific weaknesses, thus lacking explanatory power for real-world ASR model outputs. This deficiency hinders the establishment of clear directions for model improvement. To this end, datasets that aim to enhance the explanatory power of ASR evaluations by considering noisy environments or speaker characteristics have been published recently (Sikasote and Anastasopoulos, 2022; Lakomkin et al., 2019; Gong et al., 2022; Dai et al., 2022). However, these datasets still focus on accuracy and provide a limited set of error types, thereby leading to insufficiency in diagnosing specific weaknesses within ASR models.

Therefore, we introduce the novel **K**orean **E**rror Explainable **B**enchmark Dataset for **A**SR and **P**ostprocessing (**KEBAP**)[2]. It encompasses speech-level distraction-based resources and text-level error types relevant to real-world ASR applications. These diverse error types of **KEBAP** can lead to improved explanatory capability compared to the previous examination methods, as illustrated in Figure 1. In particular, speech-level noise types are bifurcated into two categories: noisy environments and speaker characteristics, comprising 37 distinct types. Additionally, **KEBAP** includes 13 types of textual errors pertinent in ASR contexts. The dataset stands out in its authenticity since all speech samples are recorded by human speakers, and background noises are derived from real-world environments. Also, We annotate the difficulty levels to all types, enhancing the interpretability of the ASR model.

We employ **KEBAP** to conduct an empirical analysis of the correlations between speech-level noise types and textual error types. Moreover, leveraging ChatGPT (OpenAI-Blog, 2022), we explore the potential of language models in discovering the vulnerabilities of ASR models. Our observations highlight **KEBAP**'s significant interpretability of ASR model diagnostics and shed light on the pressing need for research on diagnostic tasks for ASR systems. Our work sets the stage for more real-world-oriented evaluations of ASR systems and can contribute to the advancements in this domain.

## 2 KEBAP

### 2.1 Why KEBAP?

In the real-world scenario, mitigating the trade-off between recognition accuracy and user readability is crucial. To address this, we propose **KEBAP**, emphasizing the importance of considering both aspects. A detailed explanation is as follows. Firstly, in real-world speech recognition, it is essential to consider the accuracy of model and end-user satisfaction simultaneously. To facilitate this, we propose to map the accuracy of the ASR model to 'speech-level noises' and user readability to 'text-level errors' to mitigate this inherent trade-off. From the perspective of the accuracy of the ASR model, it should output the recognition results 'as heard,' regardless of the quality of the user-provided input. Conversely, from the standpoint of the end-user receiving the result, satisfaction increases when the output is presented in a refined state, despite any errors in the initial input. For instance, if a speaker stammers during their speech, the ASR model would likely deem its output more accurate if it recognizes and outputs all the words uttered. However, this would likely result in lower readability from the user's perspective.

In addition, previous research lacks an adequate number of error types for a detailed diagnosis. Since benchmarks measure performance with quantitative metrics, it is crucial to subdivide characteristics for a more detailed diagnosis. In industry contexts, communication between model and service teams is critical. When there's an issue with the model, clear criteria for the data flywheel significantly facilitate communication. That is, distinguishing the error type criteria for speech- and text-level aids in detailed diagnosis for model improvement. However, conventional benchmark datasets lack sufficient error types for detailed model analysis, leading to extensive usage of human evaluation in real-world settings. Humans can cope using commonsense, even if the criteria are unclear, but existing benchmarks with limited error types fall short. Hence, to solve the explainability issue, we must define error type criteria that consider both the speech- and text-level and create benchmarks to achieve human-level explainability.

To enhance the explanatory power of the validation process for ASR models, we define errors

---

[2]Our **KEBAP** dataset is publicly available at `https://github.com/seonminkoo/KEBAP`

| Noise Type | | | Description |
|---|---|---|---|
| Noisy environment | Home appliances | Washer/dryer machine | Difficulty in recognition due to ambient electrical appliance noise. |
| | | Vacuum cleaner | |
| | Individual transportation | Motorcycle | Difficulty in recognition due to surrounding individual transportation noise. |
| | | Siren | |
| | | Honk | |
| | Street | Road side | Difficulty in recognition due to the surrounding street noise. |
| | | Crowd | |
| | Cafe/restaurant | Conversation | Challenges in perception due to the noise in cafes/restaurants. |
| | | Non-conversation | |
| | Market/shopping mall | Traditional market | Difficulties in perception caused by the noise in markets/shopping malls. |
| | | Shopping mall | |
| | Public transportation | Subway platform | Difficulty in recognition due to surrounding public transportation noise. |
| | | Inside the subway | |
| | | Inside the train (STR/KTX) | |
| | | Inside the bus | |
| | Terminal | Train terminal waiting room | Challenges in perception due to the noise at terminals. |
| | | Bus terminal waiting room | |
| | Construction site | Outdoor construction site | Difficulties in perception caused by the noise at construction sites. |
| | | Indoor construction site | |
| | Factory | processing process | Difficulties in perception caused by the noise in factories. |
| | | Assembly process | |
| | Nature ambient | Sound of rain | Challenges in perception due to natural ambient noise. |
| | | Sound of the waves | |
| | Etc. | Artificial mechanical sound | In cases where external noise is present, although not falling into the aforementioned categories. |
| Characteristics of interlocutor | Pause (silent) | | When there is a presence of pauses between syllables in speech that has not yet concluded. |
| | Filled pause | | When habitual sounds are inserted during moments of silence or break time. |
| | Interjection | | When phrases or longer segments are inserted regardless of their relevance to the intended content being expressed. |
| | Parenthetical | | When grammatically acceptable sentences are inserted without conveying specific meaning or significance. |
| | Unfinished interlocutor | | When speech is terminated without concluding the sentence. |
| | Word repetition | | Repeating the same word or phrase in succession during speech. |
| | Syllable repetition | | Repeating the same syllable in succession during speech. |
| | Phoneme repetition | | Repeating the same phoneme in succession during speech. |
| | Sustained | | When elongating certain parts of words within a sentence during speech. |
| | Hyperfluency | | When excessively verbose speech is employed. |
| | Mutter | | When muttering with an unclear demeanor. |
| | Dynamic error | | When syllabic intonation is inappropriate for the intended speech purpose or difficult for human-level comprehension. |
| | Speaking rate | | When speech rate is excessively fast, making it difficult for human-level comprehension. |

Table 1: Proposed novel speech-level noise type classification criteria for **KEBAP**

at speech- and text-level and propose **KEBAP**, a benchmark that considers various potential error-prone environments in real-world scenarios

## 2.2 Speech-Level Noise Type

Error types at the speech-level refer to factors that trigger inaccuracies in speech recognition situations. For example, identical utterances may be challenging to recognize due to background noise (Sikasote and Anastasopoulos, 2022). Additionally, even in quiet environments, individuals do not consistently articulate perfect sentences and each speaker has unique characteristics that may negatively influence speech recognition (Gong et al., 2022).

Table 1 illustrates the speech-level error type classification criteria considering these characteristics. The speech-level error types allow the classification of two main categories (noisy environment and characteristics of interlocutor) and more detailed error types, with 24 sub-types for noise error

and 13 for speaker characteristics.

Considering environments inundated with noise, it does not represent a quiet recording situation but rather a condition intertwined with noise. Real-world scenarios frequently involve inputs replete with ambient noise (Sikasote and Anastasopoulos, 2022). Reflecting on these practical situations where voice interface applications and devices are deployed, we propose an enhanced categorization scheme that closely follows the classification in the AI-HUB's noisy environment speech recognition dataset [3] which are representative Korean data platform. We divide the noisy environment errors into 11 nuanced subcategories, including **home appliances**, where recognition is impaired due to surrounding appliance noise; **individual transportation**, which includes instances with ambient transportation noise; **street**, covering situations with disruptive street noise; **cafe/restaurant**, addressing cases with the cafe or restaurant ambient noise;

---

[3] https://www.aihub.or.kr/

| Category | | | Description |
|---|---|---|---|
| **Level A** | **Level B** | **Level C** | |
| Spacing | - | - | Violating the spacing rules. |
| Punctuation | - | - | Punctuation marks are not attached in Korean sentences or are attached in the wrong. |
| Numerical | - | - | Cardinal number indicating quantity and the ordinal number indicating the order are in error |
| Spelling and Grammatical | Remove | - | Some words are not recognized, or endings or suffixes are omitted. |
| | Addition | - | Same word is repeated, or an unused postposition or ending is added. |
| | Replace | - | Word is replaced by another word. |
| | Separation | - | Separating consonants and vowels in characters. |
| | Foreign word conversion | - | Instances of incorrect conversion of syllables between English and Korean, as well as writing spellings according to pronunciation, have been observed. |
| | Spelling | G2P | Writing spellings according to pronunciation. |
| | | CVC | Spelling error in non-speaking alphabet units. |
| | Post-position | - | Instances of inconsistent or missing post-position usage in target utterances. |
| | Syntax | - | Cases of grammatically accurate yet interpretatively ambiguous meanings. |
| | Neologism | - | Instances of the discrepancy between target and its similarity in meaning, pronunciation, and absence in Korean lexicon. |

Table 2: Proposed text-level error type classification criteria for **KEBAP**. G2P and CVC indicate Grapheme-to-phoneme and Consonant vowel conversion, respectivity

**market/shopping mall**, indicating instances with market or shopping mall noise; **public transportation**, comprising cases with subway or bus noise; **terminal**, reflecting instances with terminal noise; **construction site**, for cases hindered by construction site noise; **factory**, indicating instances with factory noise; **nature ambient**, for cases disturbed by natural sounds. Lastly, we include an **etc.** category for instances where recognition is affected by external noise types not encompassed in the previous categories.

Considering speaker characteristics, recognition can be hampered due to the individual traits of the recorder. Inspired by studies on idiolectal elements in the field of psycholinguistics (Ha and Sim, 2008; Shin et al., 2005), we propose a nuanced categorization comprising 13 detailed subcategories. The details description of the subcategories are described in Appendix B.

## 2.3 Text-Level Error Type

Text-level error types refer to issues that emerge in speech recognition results and must be addressed by post-processing. Since the output of the speech recognizer serves as the input for downstream tasks, it is one of the most significant factors influencing end-user satisfaction. By improving the performance of downstream tasks through quality input and diagnosing the performance of post-processing models through detailed error types, it is possible to enhance end-user satisfaction.

Existing datasets that detail error types, such as grammatical error correction (GEC) datasets, do not consider speech recognition situations (Koo et al., 2022; Yoon et al., 2022). Therefore, we re-

configure the Korean GEC dataset, K-NCT, to suit speech recognition situations. The existing K-NCT dataset includes errors that only occur at the text-level and not in speech situations (Koo et al., 2022). Hence, errors that do not have vocal characteristics are removed.

Table 2 illustrates the text-level error type classification criteria considering speech recognition situations, including 13 text-level errors that can occur in speech recognition situations. Detailed explanations for each text-level error type can be found in Appendix C

## 2.4 KEBAP Construction Process

In this work, we propose a comprehensive data construction guideline for the ASR and ASRP dataset, grounded in the application of a GEC dataset. Our methodology encompasses build text-level error corpus, speech recording, noise synthesis, and difficulty annotation. For the efficiency of the task, we choose the 'consensus labeling' method (Tang and Lease, 2011), in which a human overseer, who possesses an elevated degree of task completion, serves as a quality controller. During the progression of the task, any outcomes that do not conform to the established guidelines are promptly dismissed and subsequently reconstructed.

**Step 1: Build Text-Level Error Corpus** In this study, we employ a human-curated GEC dataset, which encompasses various text-level error types (Koo et al., 2022). Considering the inapplicability of the standard GEC benchmark dataset in a speech recognition setting, we selectively compose a text-level error types dataset by human evaluation.

We assess whether the given error types are valid or invalid in the context of speech recognition situations by human evaluators. Invalid types are filtered out, and the type structure is reconfigured In particular, we extract 13 categories that resonate with speech recognition scenarios (e.g., honorific colloquial expression) and reorganize their hierarchy for ease of labeling. Consequently, our refined dataset includes data reflecting 13 error types relevant to speech recognition contexts.

Subsequently, we authenticate the quality of the filtered dataset focusing on the alignment between labels and text, and the inclusion of text-level errors with a specific consideration of the speech recognition context. Validation processes proceed with a human supervisor, priorly trained with each error type. Evaluators are presented with an erroneous sentence, its correct counterpart, and a specified error type with the corresponding error span indicated. They are then tasked with assessing whether the sentence contains the presented error types. Sentences deemed to be incorrect are appropriately amended. This procedural framework ensures the generation of a high-quality dataset.

**Step 2: Speech Recording**  In the second phase, we request that recording participants incorporate characteristics of interlocutor errors into their recordings by presenting them with speech-level errors and transcription relevant to the respective error types. At most 3 error types are presented, which could include an instance of 'no error type', indicating clean data. The placement of the error within the sentence is non-specific, with the ensurance that it includes only the errors specified. The recording environment should be ensured to be quiet without background noise. Each recorder is instructed to speak as naturally as possible, emulating their speech patterns when interacting with a voice interface application in real-world scenarios. After completing the recording, participants have the opportunity to listen to their own voice, and if they determine that the speech does not meet the criteria, they can re-record it. Participants are required to go through the process of listening to their recorded speech in order to complete the recording task. The detailed information about the workers can be found in Appendix D.

**Step 3: Synthesis of Background Noise**  In the next stage, we incorporate background noise into the recording to reflect the noise environment er-

ror in the proposed speech-level. The background noise used for this integration is derived directly from recordings of the identified environments. We ensure that the collected noise spans a duration longer than that of the recording file, fostering noise diversity. To mimic real-world situations, we conduct both single and multiple noise syntheses while filtering out instances that are unlikely to co-occur. During noise synthesis, the noise is integrated as though it is ambient background noise, designed to be audible at the onset of the voice file. Noise is composited into the recording by randomly excising sections, thus ensuring variation within sounds, even when they are categorized under the same noise type.

**Step 4: Difficulty Annotation**  Difficult data for ASR models refers to data that is not frequently encountered in the training data and is imbalanced, varying depending on the user (Aleksic et al., 2015a). Therefore, we annotate the difficulty of the data to enable a detailed assessment of the model's coverage ability. To this end, we employ a framework that distinguishes between utterances considered easier for ASR and those deemed harder or more noisy for ASR (Breiner et al., 2022). We extend this framework to include the tagging of difficulty using a Likert scale by human annotators. Humans listen to audio file and select score based on evaluation criteria. We ask humans, 'How difficult is it to recognize the presented speech accurately as the same as the transcript?' Scores range from 1 (very easy) to 5 (very difficult). Three evaluators assess each audio file, and the average score is selected as the difficulty level of the data. This allows for a detailed analysis of the model's performance. The details of construction process described in Appendix D.

## 3 KEBAP Analysis

### 3.1 Text-level Distribution

We filter out cases that cannot occur in speech recognition situations, such as typing language errors caused by keyboard language switching, in the GEC dataset. After the filtering process, the text-level distribution is shown in Table 3. It includes 2,478 instances of errors and correct sentences, including text-level errors. The statistical information for the text is provided in Table 4

| Category | | | Count (%) |
|---|---|---|---|
| **Level A** | **Level B** | **Level C** | |
| Spacing | - | - | 514 (14.62) |
| Punctuation | - | - | 505 (14.37) |
| Numerical | - | - | 500 (14.22) |
| | Remove | - | 122 (3.47) |
| | Addition | - | 104 (2.96) |
| | Replace | | 483 (13.74) |
| Spelling | Separation | - | 94 (2.67) |
| and | Foreign word conversion | - | 193 (5.49) |
| Grammatical | Spelling | G2P | 195 (5.55) |
| | | CVC | 534 (15.19) |
| | Post-position | - | 90 (2.56) |
| | Syntax | - | 71 (2.02) |
| | Neologism | - | 110 (3.13) |
| | Total | | 3515 (100) |

Table 3: Statistics of labels in text level category of **KE-BAP**. Here, the lowest level of data granularity is the category attribute in Level C. G2P and CVC are indicated as Grapheme-to-phoneme and consonant vowel conversion, respectively.

| KEBAP | Test | |
|---|---|---|
| | **Error sentence** | **Correct sentence** |
| # of sents | 2,478 | 2,478 |
| # of tokens | 107,411 | 107,209 |
| # of words | 25,772 | 26,250 |
| | | |
| avg of SL △ | 43.35 | 43.26 |
| avg of WS | 10.40 | 10.59 |
| avg of SS | 9.40 | 9.59 |

Table 4: Statistics of our **KEBAP** dataset. # of sents/tokens/words: number of sentences/tokens/words; △ avg of SL/WS/SS: average of sentence length/words/spaces per sentence.

## 3.2 Speech-level Distribution

**KEBAP** consists of a total of 2,478 speech files, transcriptions, and speech-level noise types. Figure 2-(a) illustrates the data distribution for speech-level. **KEBAP** is composed of a total of 24,021.82 seconds of speech. It includes an average speech duration of 9.69 seconds, with the shortest file being 3.8 seconds and the longest file being 27.68 seconds. Although transcription sentences are composed of single sentences, their lengths can vary depending on speaker characteristics, such as pauses (silent) or mutters, even for sentences of the same length. This allows for the inclusion of speech files of varying durations, covering the characteristics of diverse users who use ASR systems.

## 3.3 Difficulty Distribution

Overall, the average Krippendorff's $\alpha$ for inter-annotator agreement of each annotation level is 0.476. The label distribution of the collected data is

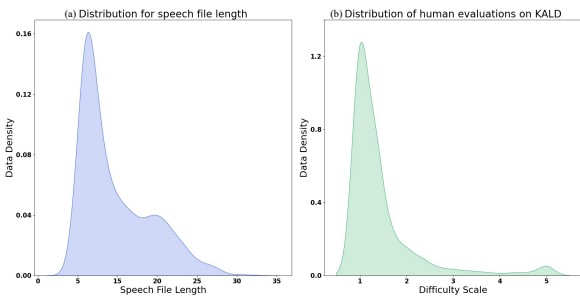

Figure 2: Data distribution for speech-level. (a) indicates distribution for the speech file length. (b) indicates the difficulty distribution of human evaluation.

shown in Figure 2-(b). To more accurately diagnose the model's capability, we enhance its interpretability by tagging the difficulty level of the data. Since the perceived difficulty of the same data may vary among individuals, we determine the difficulty of each data based on the average difficulty annotation provided by three evaluators. The difficulty ratings for the data are generally concentrated between 1 and 2, but there is also a significant presence of ratings at 5. This indicates that the dataset includes a range of difficulty levels, which we believe will be beneficial for assessing the performance of ASR models.

## 3.4 Category Distribution

Each data includes single or multiple speech-level characteristics. Figure 3 shows the distribution of each category of speech-level. The speech-level errors can be broadly classified into two main categories: noisy environment and characteristics of the interlocutor. Our dataset encompasses various combinations of characteristics within each category and also includes cross-category combinations, providing a diverse range of error types. Noisy environment and characteristics of the interlocutor represent mutually exclusive types, while co-occurrence indicates cases where two characteristics occur simultaneously. When considering only noisy environments, 1 and 2 characteristics account for 40.02% each, and 3 characteristics account for 19.96%.

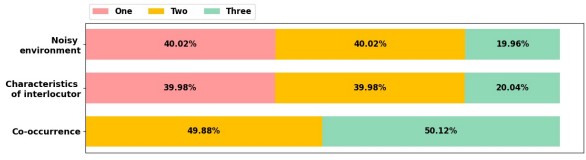

Figure 3: Distribution of each category of speech-level.

Noisy environments that cannot occur simultaneously are not included. When considering only the characteristics of the interlocutor, 1 and 2 characteristics account for 39.98% each, and 3 characteristics account for 20.04%. Co-occurrence occurs in 49.88% of cases for 2 characteristics and 50.12% of cases for 3 characteristics. This demonstrates the presence of a diverse range of error levels, both in terms of types and quantities. Actual workers recorded the data, and the background noise was collected directly from real-world environments, ensuring high quality. There is no synthetic audio involved in the recordings.

| | WER | | | CER | | |
|---|---|---|---|---|---|---|
| | Easy | Medium | Hard | Easy | Medium | Hard |
| Google ASR | 0.47 | 0.63 | 0.93 | 0.21 | 0.34 | 0.69 |
| Clova ASR | 0.53 | 0.67 | 0.94 | 0.2 | 0.35 | 0.73 |
| Whisper | 0.48 | 0.67 | 0.92 | 0.23 | 0.35 | 0.65 |

Table 5: Evaluation results of ASR commercialization systems and publicly available model (Radford et al., 2023). Word Error Rate (WER) and Character Error Rate (CER) indicate better performance as their values decrease.

## 4 Efficacy Validation for KEBAP

In this section, we assess the specific capabilities of commercialized ASR models using **KEBAP**. To achieve this, we conduct a detailed correlation analysis of commercialized systems such as Google Cloud Speech-to-Text (Aleksic et al., 2015b) and CLOVA Speech (Chung, 2019). We examine the correlation between speech-level noise types and text-level errors, aiming for a granular understanding. We comprehensively validate the model's capabilities by considering both speech- and text-level aspects. We verify whether the LLM possesses the necessary qualities as a diagnostic model through the error type classification task.

### 4.1 Analysis of Correlation in ASR models

Table 5 shows the evaluation results of ASR models. Based on conventional evaluation metrics such as WER (Woodard and Nelson) and CER (Morris et al., 2004), we observe similar performance between the two ASR models. However, even though the quantitative evaluation results may be similar, it does not necessarily mean that the qualitative aspects of the ASR model's outputs are also similar. This makes it challenging to identify the specific

weaknesses of the model, hindering the establishment of directions for model improvement. To enhance interpretability, we analyze the tendency of text-level error propagation at the speech level for ASR model. To clearly understand the impact of each speech-level category on the text-level, we sample data that includes a single speech-level noise type. The results of the ASR model are labeled by humans trained in explanations and examples of text-level errors.

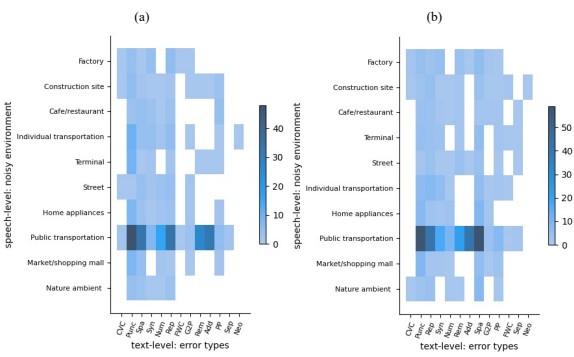

Figure 4: Correlation distribution between speech-level: noisy environment and text-level in Google(a) and Clova(b). Spa/Punc/Num represent spacing, punctuation, and numerical, respectively. Rem/Add/Rep/Sep indicate remove, addition, replace, and separation, respectively. FWC/G2P/CVC correspond to foreign word conversion, grapheme-to-phoneme, and consonant vowel conversion, respectively. PP/Syn/Neo signify post-position, syntax, and neologism, respectively.

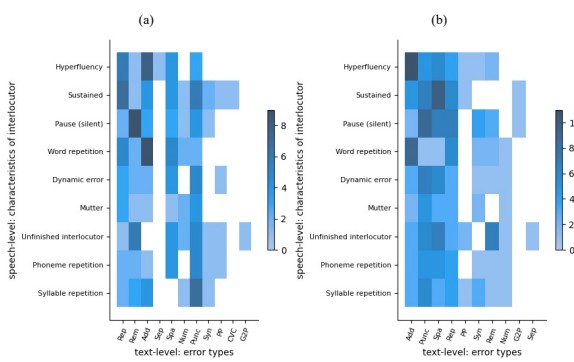

Figure 5: Correlation distribution between speech-level: characteristics of interlocutor and text-level in Google(a) and Clova(b).

Figure 4 illustrates the correlation between the noisy environment at the speech level and text-level errors. In the speech-level category, we have grouped similar types together, excluding miscellaneous types. In the case of text-level punctuation errors, we consider only those instances where peri-

ods ('.') are missing in all sentences, leading to the omission of other punctuation marks such as question marks ('?') or exclamation points ('!'). This specific condition allows us to focus on scenarios where the absence of periods directly affects the presence of other punctuation marks in the transcriptions.

Both the Google and Naver ASR systems exhibit significant error propagation in the domain of public transportation. Specifically, for Google, there is a high correlation between speech-level errors and text-level errors in punctuation, spacing, and replace. On the other hand, Clova shows a strong correlation between speech-level errors and text-level errors in punctuation, spacing, and addition. Furthermore, google showed robustness in the nature ambient, but Clova showed relatively more text errors.

Figure 5 shows the correlation between speech-level characteristics of interlocutor and text-level errors. For Google, the presence of pause (silent) in speech had a significant impact on the occurrence of remove errors in the transcriptions, while word repetition contributed to the occurrence of addition errors. In the case of Clova, overall, a higher number of errors were observed compared to Google. Particularly, hyperfluency had the most significant impact on the occurrence of addition errors in the transcriptions.

This analysis provides valuable insights into the correlation between speech-level noise, particularly noisy environments, and text-level errors in the Google and Clova ASR systems. The varying impact of different types of speech-level characteristics on text-level errors highlights the need for further granularity in categorizing these types. Even if models demonstrate similar performance, the individual capabilities of each model can differ. This demonstrates that **KEBAP** helps enhance the interpretability of ASR model verification [4].

## 4.2 Adequacy of Synthesized Noise

Table 6 shows the performance before and after noise synthesis. Experimental results show that for Google, the WER is 0.49, the CER is 0.23 before noise synthesis, and the WER is 0.68 and the CER is 0.41 after noise synthesis. For Clova, the WER before noise synthesis is 0.53, and the CER is 0.19,

---

[4]Additionally, **KEBAP** provides difficulty information, enabling even more fine-grained analysis. Appendix E includes the detailed analysis results based on diverse levels of difficulty.

|  | WER | | CER | |
|---|---|---|---|---|
|  | Clean | Noise | Clean | Noise |
| Google ASR | 0.49 | 0.51 | 0.23 | 0.25 |
| Clova ASR | 0.53 | 0.57 | 0.19 | 0.24 |

Table 6: Performance of commercialize systems based on the presence or absence of noise synthesis. 'Clean' and 'Noise' represent the settings before and after noise synthesis, respectively.

while the WER after noise synthesis is 0.71 and the CER is 0.43. These results are interpretable in that the resources we provide are high-quality and helpful.

## 4.3 Examination of ASR models through LLM

With recent advancements in Large Language Model (LLM) development, most tasks are converging towards LLM-based approaches. In this study, we explore the potential of using Chat-GPT (OpenAI-Blog, 2022) as a diagnostic tool for ASR results. Understanding error types is essential for verifying the models, and to measure this understanding, we perform an error type classification task. ChatGPT is utilized to classify text-level error types based on provided sentences in a few-shot setup. The specific prompt used for this experiment is listed in Appendix F.

We task ChatGPT with classifying all text-level errors occurring in the ASR results. However, as seen in the examples (please refer to Appendix F.2), it is evident that ChatGPT not only misclassifies text-level errors but also struggles more when multiple errors are present within a sentence. Although LLMs are converging towards covering various tasks, they exhibit limitations in performing diagnostic tasks for commercial systems. This indicates that while various tasks may converge with LLM, the diagnostic domain for the proposed model is far from convergence with LLMs, highlighting the need for further research.

## 5 Conclusion

In the real-world, ASR results involve a trade-off between recognition accuracy and user readability, thus requiring a balanced consideration of these factors. To provide guidance for improving model performance, it is necessary to enhance interpretability, which entails considering both speech-level accuracy and text-level user readability. To this end, we propose **K**orean **E**rror Explainable **B**enchmark

Dataset for **A**SR and **P**ost-processing (**KEBAP**) for diagnosing and validating models by segmenting error types while considering both speech- and text-level. To facilitate the construction process, we utilize a GEC dataset that includes text-level errors and structure the process into validation, recording, synthesis of background noise, and difficulty tagging stages, employing consensus labeling within each stage to enhance the efficiency and quality of the task. We performed a detailed diagnostic analysis of the commercialization systems using **KEBAP**. Furthermore, the proposed task falls into a domain that is challenging for ChatGPT to cover, and it indicates the need for further research to achieve a closer approximation to real-world diagnostics. We demonstrated that **KEBAP** contributes to enhancing the interpretability of the model's weaknesses.

## Limitations

This study has the limitation of only building data for the Korean language. Additionally, as this paper proposes a new task, it was not able to conduct extensive quantitative analyses by comparing it with existing models, which remains a limitation. However, this paper made a contribution by proposing new data and tasks and making them publicly available.

## Ethics Statement

We discuss the main ethical considerations of **KEBAP** benchmark we presented: (1) Privacy. **KEBAP** benchmark is constructed to acquire factual dataset, and does not contain privacy issues. (2) Human evaluation. During data evaluation process, we paid human workers the legal wage determined by the average time of evaluation and local labor compensation standards. We also guided them to take a rest when they are in a state of fatigue during work. (3) Potential problems. While principled measures are taken to ensure the quality of the dataset, there might still be potential problems with the dataset quality.

## Acknowledgements

This research was supported by the MSIT(Ministry of Science and ICT), Korea, under the ITRC(Information Technology Research Center) support program(IITP-2023-2018-0-01405) supervised by the IITP(Institute for Information & Communications Technology Planning & Evaluation). This work was supported by Institute of Information & communications Technology Planning & Evaluation(IITP) grant funded by the Korea government(MSIT) (No. 2020-0-00368, A Neural-Symbolic Model for Knowledge Acquisition and Inference Techniques). This work was supported by Institute for Information & communications Technology Planning & Evaluation(IITP) grant funded by the Korea government(MSIT) (No. 2022-0-00369, (Part 4) Development of AI Technology to support Expert Decision-making that can Explain the Reasons/Grounds for Judgment Results based on Expert Knowledge)

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

# A Related Works and Background

**Post-processing Model** Post-processing serves an important role in quality enhancement across various fields by modifying the distorted output into appropriate statements. For instance, in the field of optical character recognition (OCR), conventional approaches such as manual, lexical, and statistical methods have been used (Evershed and Fitch, 2014; Nguyen et al., 2018). More recently, language models like BERT have been employed for error detection in tasks like named entity recognition (NER) and are performed through character-level machine translation (Nguyen et al., 2020).

As another field, machine translation (MT) often utilizes the following methods. Post-processing research is being carried out in automatic post-editing (APE) to improve translation quality by adopting transfer learning (Correia and Martins, 2019). Concurrently, in the grammatical error correction (GEC) field, transformers and the copy mechanism are used to correct spelling and grammatical errors in MT results (Lee et al., 2021). Studies that define error types to construct test sets or utilize an automatic grammatical error annotation system to create datasets also exist to improve Korean GEC studies (Koo et al., 2022; Yoon et al., 2022). Likewise, the study on post-processing is actively explored in a wide range of fields and holds significance in terms of enhancing the quality of output results. This can also be of significant importance in the field of Automatic speech recognition (ASR), which is discussed in the following section.

**ASR Post-Processing Model** ASR post-processing (ASRP) involves the detection and correction of errors in the output of an ASR, distinguishing it from simple error correction in that it considers user-friendliness as an additional aspect. This approach can improve the final quality of statements without modifying the ASR system structure. For instance, in specialized fields like the medical domain, attempts have been made to eliminate punctuation errors in ASR through post-processing (Mani et al., 2020a). Prior research has primarily focused on providing information that allows humans to manually rectify erroneous segments, proposing alternative words for correction or creating an environment conducive to modification (Suhm et al., 2001; Feng and Sears, 2004). External information, such as word alternative hypothesis, noisy context, and accurate context, is provided to assist in post-processing for error correction (Shi and Zhou, 2011). In particular, Bassil and Semaan (2012) use the N-gram dataset for ASR errors to detect and correct errors automatically. Models such as LSTM-based or Transformer-based sequence-to-sequence architectures are adopted to correct the speech recognition results while considering the semantics and spelling (Guo et al., 2019; Hrinchuk et al., 2020).

Recent studies strive to improve ASRP performance by utilizing the results derived from ASR. Gekhman et al. (2022a) introduce the ASR confidence embedding (ACE) layer to the encoder of the ASR model to jointly encode the confidence scores and transcribed text into a contextualized representation. To mitigate the time and cost-related challenges associated with the parallel data required for training, Park et al. (2021) employ Text-to-speech (TTS) and Speech-to-text (STT) technologies to construct parallel data.

**ASR dataset** The availability of suitable datasets is imperative for the active progression of ASRP. Previously, post-processing studies have been conducted with ASR datasets. Panayotov et al. (2015) organize the two labels in the ASR dataset that denote the quality of speech recognition, classified into 'clean' and 'other' categories, providing valuable assistance in the analysis. Ardila et al. (2020) construct comprehensive ASR dataset that includes demographic metadata such as age, sex, and accent to provide a wider representation.

Transcription hypotheses obtained by decoding audio data using an ASR model are used to align hypothesis words with the reference (correct) transcription. The process of labeling errors and non-errors is facilitated by employing the minimum edit distance (Gekhman et al., 2022b). In the context of Chinese language datasets, a significant dataset is available for speech recognition systems, labeled with audio devices and recording environments (Bu et al., 2017). Gekhman et al. (2022b) build a dataset by aligning hypothesis words with the reference (correct) transcription through a transcription hypothesis obtained by decoding audio data with an ASR model and labeling errors and nonerrors using minimum edit distance. In the context of Chinese, a large-scale dataset is available for speech recognition systems labeled with audio device information and recording environments (Bu et al., 2017).

To mitigate the problem of insufficient training

data, methodologies that synthesize data via data augmentation methods have been proposed (Liao et al., 2022). However, the overall quality of the data is more crucial than the size. Specifically, the detailed datasets that consider both speech- and text-level like the real world are absent. Consequently, we aim to construct the ASR Post-Processing dataset, which contemplates audio- and text-level for the first time.

## B  Description of Speech-Level Noise Type

**Pause (silent)** category captures instances where silence intervenes mid-utterance before completion—for instance, when 'I am eating' is articulated as 'I am... eating'. **Filled pause** represents cases characterized by the habitual insertion of filler sounds during pauses, as in utterances supplemented by sounds such as 'um... uh... so I'. **Interjection** category encompasses instances where one or more words or phrases irrelevant to the intended message are interjected, evident in utterances like 'Okay I see, but you know'. **Parenthetical** category includes instances where grammatically correct, but semantically neutral phrases are inserted—for instance, utterances incorporating phrases such as 'you know' and 'I mean'. **Unfinished interlocutor** category denotes cases where the utterance concludes prematurely—for instance, when 'I am eating' is truncated to 'I am...'. **Word repetition** category signifies instances where the same word is iterated, as in saying 'Hello' as 'Hello Hello'. **Syllable repetition** category characterizes cases where the same syllable is iterated—for instance, when 'Hello' is articulated as 'He-hello'. **Phoneme repetition** category encapsulates instances where the same phoneme is repeated, such as saying 'Hello' as 'Hel-llo'. **Sustained** category accounts for instances where part of an utterance is elongated, exemplified in 'Is that so—right?'. **Hyperfluency** category represents instances of excessive verbosity. **Mutter** category includes cases where utterances are murmured in an indistinct manner, as in 'That.. is.. like that...'. **Dynamic error** category encompasses instances where syllable articulation strength is incongruous with the intended utterance, or instances that are challenging to comprehend at the human-level. Finally, **speaking rate** category accounts for instances where rapid speech pace hinders comprehension at a human-level.

## C  Description of Text-Level Error Type

**Spacing** encapsulates instances contravening standard spacing conventions. **Punctuation** entails cases where punctuation is omitted or misapplied in Korean sentences—for instance, when 'Can I teach?' is interpreted as 'Can I teach.' **Numerical** encompasses cases where number conversion fails, such as when 'Ahead of the three-month schedule' is interpreted as 'Bill 2, 3-month schedule'.

**Spelling and Grammar** consists of ten detailed subcategories. **Remove** designates cases where some word components are not recognized, or endings or particles are missing—for example, when 'The champion is in the final' is misinterpreted as 'Champion final'. **Addition** involves cases where the same word is repeated or unutilized particles or endings are appended. For instance, when 'World's fruits, fish, and meat' is interpreted as 'World's world's fruits, fish, and meat'. **Replace** refers to instances where one word is substituted with another—for example, when 'Apply the filter.' is interpreted as 'Wear the pizza'. **Separation** refers to instances where consonants and vowels in the target utterance are separated, exemplified when 'The discount applies as it is.' is interpreted as 'Discount app - lise as it is.'. **Foreign word conversion** refers to cases where words deviate from standard foreign word pronunciation or some syllables are incorrectly converted from English to Korean or vice versa. For example, when 'Brazil's Samba Festival' is interpreted as 'Brazil's SsamBap Festival,' or 'I prefer to use ATM.' is interpreted as 'I prefer to use hm.'.

**Spelling** is bifurcated into two types: Grapheme-to-Phoneme (G2P) and Consonant vowel conversion. **G2P** pertains to instances where a character is recognized per its pronunciation. **Consonant vowel conversion** refers to instances where phonemic units are incorrectly spelled. **Post-position** refers to cases where different particles are used or omitted—for example, when 'Ordinary high school students' is interpreted as 'Ordinary at high school students.' **Syntax** involves cases where the grammatical interpretation remains valid, but the semantic interpretation varies. Finally, **neologism** refers to cases where the target word and its meaning and pronunciation are dissimilar and are not included in Korean vocabulary.

## D Human Annotation

### D.1 Crowd-sourcing and Compensation

We recruited individuals who are native speakers of Korean and selectively hired candidates suitable for the task through validation questions. Every employee has been fairly remunerated at least a rate of 140 KRW per task. It is expected that each worker will complete 2-3 questions within a minute, guaranteeing a minimum compensation of 16,800 KRW per hour. Comparatively, the minimum hourly wage in South Korea for 2023 is 9,620 KRW. The guidelines for annotation and the user interface are illustrated in Figure 6 and Figure 7.

### D.2 Annotation Guidelines and Interface

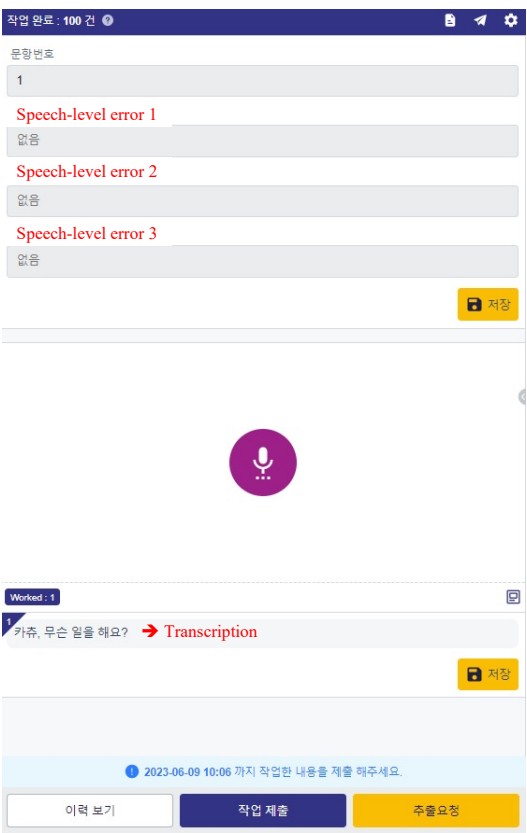

Figure 6: **Speech recording setup.**

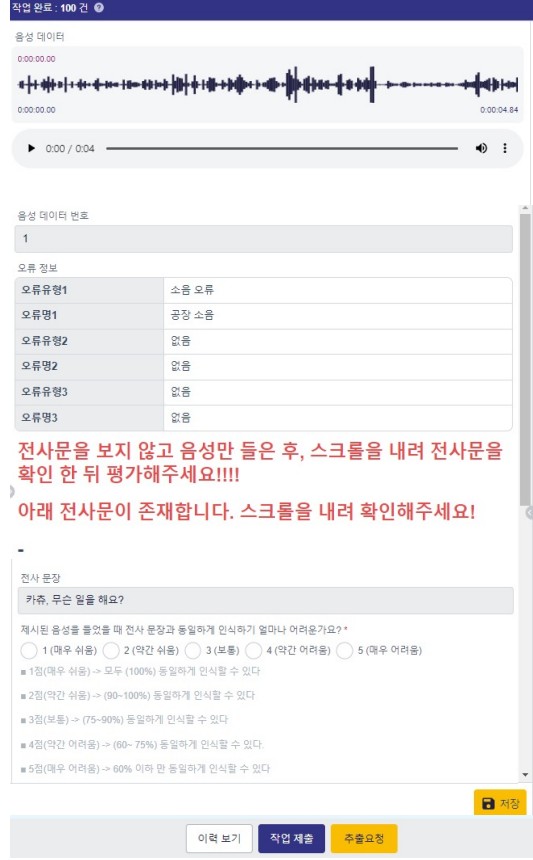

Figure 7: **Difficulty annotation setup.** Q: Please evaluate the level of difficulty in accurately transcribing the speech to match the given transcript.

### D.3 Annotation Demographics

The detailed demographic information is provided in Table 7

| | Gender |
|---|---|
| **Male** | 186 (7.50%) |
| **Female** | 2292 (92.49%) |
| | **Age** |
| **20-29** | 479 (19.33%) |
| **30-39** | 1318 (53.19%) |
| **40-49** | 681 (27.48%) |
| | **Nationality** |
| **Korea** | 2478 (100%) |

Table 7: Demographics of the crowd workers involved in the composition of the data.

## E Analysis of Correlation by Difficulty Levels

We believe that providing difficulty information facilitates the analysis of weaknesses in ASR models. We extracted an equal number of samples for each difficulty level and analyzed them. Figure 8, Figure 9, and Figure 10 show the correlation between the noisy environment at the speech level and text-level errors in diverse difficulty settings. Figure 11, Figure 12, and Figure 13 illustrate the correlation between speech-level characteristics of interlocutor and text-level errors in diverse difficulty settings. Spa/Punc/Num represent spacing, punctuation, and numerical, respectively. Rem/Add/Rep/Sep indicate remove, addition, replace, and separation, respectively. FWC/G2P/CVC correspond to foreign word conversion, grapheme-to-phoneme, and consonant vowel conversion, respectively. PP/Syn/Neo signify post-position, syntax, and neologism, respectively.

Analyzing the details based on different difficulty levels can be employed to enhance the interpretability of the ASR model. For example, in the case of Google, experimental results show that the correlation from 'Terminal' speech-level type to 'Punctuation' text-level type is strong for easy level, 'Construction site' speech-level type to 'Addition' text-level type for medium level, and 'Terminal' speech-level type to 'Replace' or 'Remove' text-level type for hard level. For Clova, the tendency of 'Replace' text-level type in 'Individual transportation' speech-level type is strongest at easy level, and it is strongly related to 'Syntax' and 'Replace' text-level type at medium level. At the hard level, it has a strong tendency to 'Remove' and 'Syntax' text-level types.

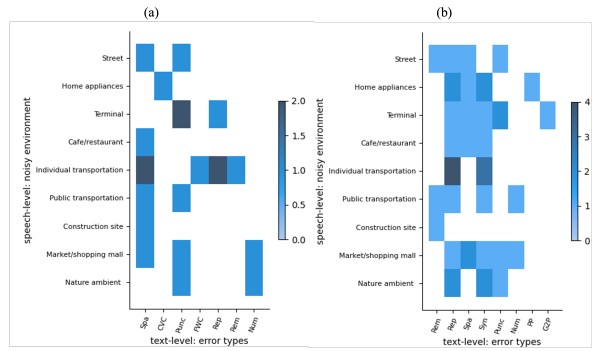

Figure 8: Correlation distribution between speech-level: noisy environment and text-level in Google(a) and Clova(b), in easy-level difficulty setting.

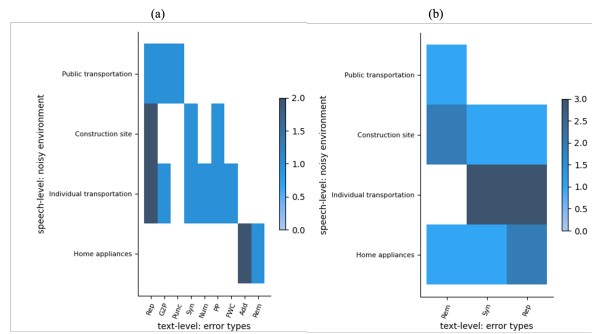

Figure 9: Correlation distribution between speech-level: noisy environment and text-level in Google(a) and Clova(b), in medium-level difficulty setting.

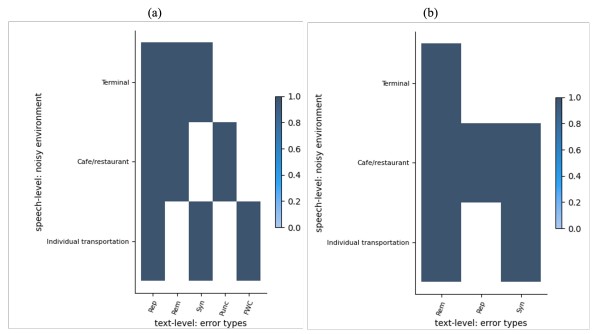

Figure 10: Correlation distribution between speech-level: noisy environment and text-level in Google(a) and Clova(b), in hard-level difficulty setting.

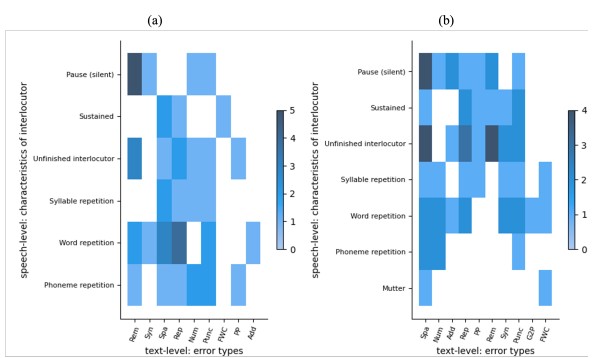

Figure 11: Correlation distribution between speech-level: characteristics of interlocutor and text-level in Google(a) and Clova(b), in easy-level difficulty setting.

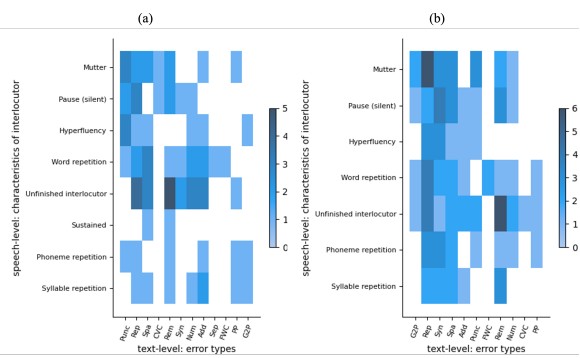

Figure 12: Correlation distribution between speech-level: characteristics of interlocutor and text-level in Google(a) and Clova(b), in medium-level difficulty setting.

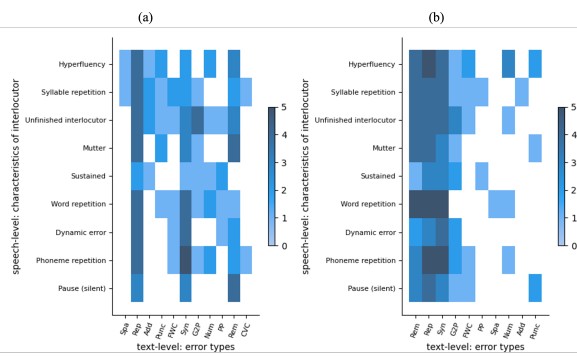

Figure 13: Correlation distribution between speech-level: characteristics of interlocutor and text-level in Google(a) and Clova(b), in hard-level difficulty setting.

# F  LLM for Validation

## F.1  Prompts of ChatGPT



**Task Description**

Your task is to classify any spelling or grammar errors within a sentence.
Always answer in Korean.

**Error Types Description**

The definition and examples are as follows:
- 띄어쓰기 오류: 띄어쓰기 규칙에 위배 되는 경우
  Example: 성공의 길을 열어줘요.
                → 성공의 길을 열어 줘요
  Result: 띄어쓰기 오류

\###
...
\###

------

**Examples**

Examples of classifying multiple
grammatical error types are as follows:
**{{examples}}**

------

**Input**

Referring to the definition and example, classify
grammatical error type that fit the given
sentences.
Example: **{{input sentence}}**
Result:



Figure 14: Error type classification prompt.

## F.2 Examples of Classified Types

| Input Sentence | STT Result | Predict Types | Target Types |
|---|---|---|---|
| 우리 자리가 생길 때까지 기다릴까요
(Shall we wait until a seat becomes available for us) | 우리 자리가 생길 때까지 기다릴까요?
(Shall we wait until a seat becomes available for us?) | No Error | Punctuation |
| 어떻게 해야 하는지만 알려 줄게 숙제 숙제는
스스로 해야지
(I will only instruct you on how to do it
The homework homework must be done by yourself.) | 어떻게 해야 하는지만 알려줄게. 숙제는
스스로 해야지.
(I will only instruct you on how to do it.
The homework must be done by yourself.) | Remove, Spacing | Addition, Spacing,
Punctuation |
| 학생은 교복을 입을 때 단정이 뭐야
(When students wear school uniforms, what does
'neatness' mean) | 학생은 교복을 입을 때 단정해 보여.
(When students wear school uniforms, the students
appear neat.) | Post-position | Syntax, Replace,
Punctuation |
| 내가 경찰이면 뭐 물어보려고 했어요
(What would I have asked if I were a police officer) | 내가 경찰이면 뭐 물어보려고 했어요?
(What would I have asked if I were a police officer?) | Spacing | Punctuation |
| 우리회사 영양제 신제품을 수입하고 싶으시다고들
(I he you want to import our company's new
nutritional supplement products.) | 우리 회사 영양제 신제품을 수입하고 싶으시다고
들었습니다.
(I heard you want to import our company's new
nutritional supplement products.) | Punctuation | Punctuation, Spacing,
Remove |

Table 8: Example of text-level error types classification in ChatGPT. STT result refers to the speech-to-text result.