# OpenReview forum: "KEBAP: Korean Error Explainable Benchmark Dataset for ASR and Post-processing"
_EMNLP/2023/Conference — EMNLP 2023 Main_

### Official Review · Reviewer_yM4D · 2023-08-04

**Soundness:** 4

**Excitement:**

5: Transformative: This paper is likely to change its subfield or computational linguistics broadly. It should be considered for a best paper award. This paper changes the current understanding of some phenomenon, shows a widely held practice to be erroneous in someway, enables a promising direction of research for a (broad or narrow) topic, or creates an exciting new technique.

**Paper Topic And Main Contributions:**

The manuscript introduces a novel benchmark dataset for assessing and identifying weakness and vulnerabilities associated with ASR systems at both speech and text level. The study identifies a key research problem in  current evaluation methods, which fails to provide information on the exact weakness of an ASR model . The proposed method introduces explainability through speech and text level errors. The speech level comprise of errors on noisy environments and speaker characteristics in 37 distinct error types, and 13 distinct text-level error types. Furthermore, the authors applied this proposed evaluation method to commercially deployed ASR systems such as Google  cloud ASR and CLOVA. The authors concluded by presenting a comparison analysis on errors for  each of these commercial systems and discussed possible limitations of their work.

**Reasons To Accept:**

1. Introducing a new and novel evaluation method for identifying and mitigating potential weakness in ASR systems.
2. Clarity in research problem, hypothesis and analysis of how the proposed method could benefit the community.
3. Easy to read and understand.
4. The proposed method is useful, as it helps identifies potential errors associated with models outside impressive scores on known metrics. It is interesting to analyse the models performance on these fine-grained error levels.

**Reasons To Reject:**

Null.

**Reproducibility:**

5: Could easily reproduce the results.

**Reviewer Confidence:**

4: Quite sure. I tried to check the important points carefully. It's unlikely, though conceivable, that I missed something that should affect my ratings.

---

> ### Author Rebuttal · Authors · 2023-08-28
>
> # Thanks for your comments!
>
> We truly appreciate your considerate understanding of our contribution and your high praise. We felt our work was worthwhile because of your thoughtful review. It motivates us to continue striving for excellence in our contributions to the academic community. Thank you again for taking the time to review my work and for providing such encouraging feedback.

---

### Official Review · Reviewer_L3vD · 2023-08-04

**Soundness:** 3

**Excitement:**

3: Ambivalent: It has merits (e.g., it reports state-of-the-art results, the idea is nice), but there are key weaknesses (e.g., it describes incremental work), and it can significantly benefit from another round of revision. However, I won't object to accepting it if my co-reviewers champion it.

**Paper Topic And Main Contributions:**

The author through this paper propose Korean Error Explainable Benchmark Dataset for ASR and Post-processing (KEBAP) as a new ASR evaluation methods. KEBAP enables comprehensive analysis of ASR systems at both speech and text levels, facilitating a more balanced assessment encompassing speech recognition accuracy and user readability. As part of the proposal KEBAP presents 37 newly defined speech-level error types, incorporates diverse noise environments, speaker characteristics categories and it also presents 13 distinct text-level error types. It ends with detailed analysis of noise categories and error types and also uses LLM for classifying error type.

**Reasons To Accept:**

1.	New benchmark dataset for ASR and post-processing evaluation in the Korean language and making it public for further research.
2.	Wide range of speech-level noise type classes.
3.	Brings in much needed explainability in ASR to forefront.
4.	Well written paper with technical soundness.


**Reasons To Reject:**

1.	Line 97 – clarify more explicitly how you annotate difficulty levels.
2.	Any bias to be addressed in human speaker recording and background noise addition?
3.	Missing point of view on Scaling to other languages.
4.	Could benefit from possible Ablation study in the setup.
5.	Missing comparison with other models and bench-marking.

**Reproducibility:**

4: Could mostly reproduce the results, but there may be some variation because of sample variance or minor variations in their interpretation of the protocol or method.

**Reviewer Confidence:**

3: Pretty sure, but there's a chance I missed something. Although I have a good feel for this area in general, I did not carefully check the paper's details, e.g., the math, experimental design, or novelty.

---

> ### Author Rebuttal · Authors · 2023-08-28
>
> # Thanks for your comments!
>
> Thanks for your constructive comments and encouraging words, and we sincerely appreciate your time in reading the paper. In the following, your comments are first stated and then followed by our point-by-point responses.
>
> > 1. Line 97 – clarify more explicitly how you annotate difficulty levels.
>
> - We asked the question, "How difficult is it to recognize the presented speech accurately as the same as the transcript?" to three human evaluators and measured their responses using a Likert scale. Specifically, we asked people to answer the question after listening to the recorded voice without looking at the transcript. Additionally, to mitigate subjectivity in scoring, we accompanied the scoring system with clear explanations of the intended significance of each score. For example, a score of 1 (very easy) signified that the speech could be recognized entirely (100%) identical to the transcript. Conversely, a score of 5 (very difficult) indicated that only 60% or less of the speech could be recognized identically. At the end of the evaluation, we adopted people's average score as the difficulty level. This method enhances the scoring process's objectivity and ensures a comprehensive understanding of the perceived difficulty in accurately recognizing speech.
>
>
> > 2. Any bias to be addressed in human speaker recording and background noise addition?
>
> - To address bias, we engaged a dedicated expert in the process who played a role in validation. During the stages of speaker recording and background noise addition, the expert ensured the accurate inclusion of provided sentences and noises and scrutinized any potential biases that could be socially problematic.
> - First, a debiasing step was conducted for the transcript validation process  (Section 2.4 - Step 1). Following the guidelines provided to our workers, any content that could raise societal concerns was subjected to rejection through consultation with the expert. We will explain this in more detail in the final copy.
> - In addition, Annotation Demographics are presented in Appendix D.3. KEBAP consists of recorders ranging in age from 20s to 40s. We also used more female recorders to include data of varying difficulty, as ASR AI models [1][2].
>
>
> [1] Investigating the Impact of Gender Representation in ASR Training Data:
> a Case Study on Librispeech (https://aclanthology.org/2021.gebnlp-1.10.pdf)
>
> [2] Effects of Talker Dialect, Gender & Race on Accuracy of Bing Speech and YouTube Automatic Captions (https://www.isca-speech.org/archive_v0/Interspeech_2017/pdfs/1746.PDF)
>
> > 3. Missing point of view on Scaling to other languags.
>
> - Our dataset is the first paper considering both recognition accuracy and user readability, and the constructed dataset is planned to be released. While we have focused on Korean due to cost considerations, our dataset's approach of fine-grained types is easily extendable to other languages using the same methodology. We are devoted to expanding our dataset to encompass other languages in future work.
>
>
> > 4. Could benefit from possible Ablation study in the setup.
>
> - Our main contribution lies in proposing and releasing a novel type of dataset from a practical perspective. For effective validation of KEBAP, we analyzed tendencies between speech- and text-level. Since we annotated the difficulty level of the data, we believe adding ablation studies based on the difficulty also would benefit. For this purpose, we extracted an equal number of samples for each difficulty level and analyzed them.
> - For example, in the case of Google, experimental results show that the correlation from ‘Terminal’ speech-level type to ‘Punctuation’ text-level type is the strongest for easy level, ‘Construction site’ speech-level type to ‘Addition’ text-level type for medium level, and ‘Terminal’ speech-level type to ‘Replace’ text-level type for hard level. We will include the entire results for the other models in the final copy.
> - In addition, we tried to measure the performance before and after noise synthesis. Experimental results show that for Google, the WER is 0.49, the CER is 0.23 before noise synthesis, and the WER is 0.68 and the CER is 0.41 after noise synthesis. For Clova, the WER before noise synthesis is 0.53, and the CER is 0.19, while the WER after noise synthesis is 0.71 and the CER is 0.43. These results are interpretable in that the resources we provide are high-quality and helpful.
>
>
> > 5. Missing comparison with other models and bench-marking.
>
> - We conducted experiments with the most typical and high-performing commercial systems in the Korean language to demonstrate the utility of our dataset from a real-world perspective. Additionally, we greatly value your constructive review, and in consideration of future research, we plan to perform additional experiments, including the Whisper model and other alternatives, as you have suggested.
> - For this purpose, we would like to share our comparison with Whisper, as the code and models of Whisper are publicly available for researchers.
>
> |||WER|||CER||
> |--- |:---: |:---: |:---: |:---: |:---: |:---: |
> ||Easy|Medium|Hard|Easy|Medium|Hard|
> |Google ASR|0.47|0.63|0.93|0.21|0.34|0.69|
> |Clova ASR|0.53|0.67|0.94|0.2|0.35|0.73|
> |Whisper|0.48|0.67|0.92|0.23|0.35|0.65|
>
> - We will include the analysis of trends between speech- and text-level in the final copy so that it can be an indicator for future researchers.
> - We will try to extend our work to a wider variety of models in the future. As this paper is approached from a practical point of view, we hope you will consider our 1) key findings of focusing on the commercial systems, 2) motivation, and 3) data release.
> - Furthermore, KEBAP emphasizes the balance between recognition accuracy and user readability. To this end, we have fine-grained noise types, enabling a detailed correlation analysis between speech- and text-level. While existing studies consider either recognition accuracy or user readability, no studies investigate a real-world-centric situation where the balance between the two aspects is necessary, like our study. For instance, while existing studies have categorized noise types for recognition accuracy, the number of defined types in our research is at least 3.7 times greater, resulting in the diagnosis of subdivided error types. We will enhance our paper by including statistical comparisons in the final copy.

---

### Official Review · Reviewer_chwF · 2023-08-09

**Soundness:** 4

**Excitement:**

3: Ambivalent: It has merits (e.g., it reports state-of-the-art results, the idea is nice), but there are key weaknesses (e.g., it describes incremental work), and it can significantly benefit from another round of revision. However, I won't object to accepting it if my co-reviewers champion it.

**Paper Topic And Main Contributions:**

This paper proposes a new dataset called Korean Error Explainable Benchmark Dataset for ASR and Post-processing (KEBAP). The proposed dataset addresses the limitations of the previous existing corpora by providing the readability component to assess the quality of the predictions. KEBAP provides diverse utterances design with different noise corruption and speaker characteristics categories. It presents 13 distinct text level error types and it appears that the dataset is more fine-grained and real-world-centric to evaluate the performance of the ASR systems.

**Reasons To Accept:**

The dataset provides many additional domain information that increase the diversity of the corpus. The new corpus also contains many additional evaluative metric that helps to assess the ASR system in more complete angles. As such, it is distinctive to existing dataset and adds value to the community when the corpus is available open-sourced. Besides, the work provides many interesting evaluations from the dataset.

**Reasons To Reject:**

While the paper provides the performance of Google and Clova networks, it will also be interesting to report and benchmark on some common existing networks that encourage academia to find a baseline/benchmark for their future research.

**Reproducibility:**

4: Could mostly reproduce the results, but there may be some variation because of sample variance or minor variations in their interpretation of the protocol or method.

**Reviewer Confidence:**

3: Pretty sure, but there's a chance I missed something. Although I have a good feel for this area in general, I did not carefully check the paper's details, e.g., the math, experimental design, or novelty.

---

> ### Author Rebuttal · Authors · 2023-08-28
>
> # Thanks for your comments!
>
> Thank you for your constructive comments and suggestions; they are exceedingly helpful in improving our paper. Our point-to-point responses to your comments are given below.
>
> > While the paper provides the performance of Google and Clova networks, it will also be interesting to report and benchmark on some common existing networks that encourage academia to find a baseline/benchmark for their future research.
>
> - We carried out experiments using the most representative and effective commercialized systems in the Korean language, aiming to demonstrate the practicality of our dataset from a real-world-centric perspective. Moreover, as you have recommended, we intend to conduct supplementary experiments encompassing the Whisper model and other potential alternatives.
> - For this purpose, we first provide a comparison with Whisper. Both the code and models of Whisper are publicly available to researchers, facilitating their utilization.
>
>
> |||WER|||CER||
> |--- |:---: |:---: |:---: |:---: |:---: |:---: |
> ||Easy|Medium|Hard|Easy|Medium|Hard|
> |Google ASR|0.47|0.63|0.93|0.21|0.34|0.69|
> |Clova ASR|0.53|0.67|0.94|0.2|0.35|0.73|
> |Whisper|0.48|0.67|0.92|0.23|0.35|0.65|
>
> - In the final version, we will incorporate the analysis between other models' speech- and text levels as a guide for future researchers.
> - We aim to expand our research scope to encompass a broader range of models. Given the practical perspective of this paper, we kindly request your consideration of our 1) key results centered on commercial systems, 2) motivation, and 3) releasing our dataset.
> - Furthermore, KEBAP emphasizes the balance between recognition accuracy and user readability. To achieve this balance, we have meticulously categorized noise types, enabling a thorough correlation analysis between speech and text levels. While some studies focus on either recognition accuracy or user readability individually, no study investigates a real-world-centric situation where the balance between the two sides is essential, like ours.
> - For example, some previous research has categorized noise types to improve recognition accuracy. However, our study has at least 3.7 times as many defined types, resulting in the diagnosis of fine-grained error types. We will include the statistical comparisons in the final version.

---

### Meta-Review · Area_Chair_uKug · 2023-09-19

**Recommendation:** 5

**Metareview:**

**Originality**

This paper is very original and proposes evaluating ASR systems in fairly novel way. It enables this evaluation by presenting a dataset with fine-grained annotations for noise types (both background and "speaker" noise).

**Significance**

The topic of explainability in ASR, which this paper is among the first to address, is very important. The resource that the authors have created can easily be used in conjunction with deletion, insertion and substitution errors to help diagnose ASR errors in the presence of specific source noises or speech characteristics. However, it does not appear that automatic classification of text errors using ChatGPT is feasible, so there may be limits the extent to which this error analysis can scale to other ASR systems.

**Clarity**

The paper is generally well structured, and most details are explained. Most reviewers agreed the paper was well-written, however, as the meta-reviewer, I did want to inform the authors of a number of grammatical, spelling or other errors that I found while reading the paper, and which I found made it slightly difficult follow at times, but which are easily fixable.

(1). "Our motivation for proposing a KEBAP, which contains both aspects, is detailed below." -- Which two aspects? It was not clear.

(2) "Since benchmarks measure performance with quantitative metrics, it is crucial **to fine-grain** characteristics for a more detailed diagnosis" -- Perhaps there was a missing word here? it is crucial to **examine**?

(3) "Hence, to solve the explainability issue, we must define **the** error type criteria that consider both the speech- and text-level and create benchmarks to achieve human-level explainability."  -- remove **the**

(4) GEC acronym in 2.3 is not defined until the subsequent section

(4) "we selectively compose **a** text-level error types dataset by human evaluation."

Step 1: Build Text-Level Error Corpus (An example would be nice for clarification)

(5) "we request the recording participants to incorporate characteristics" --> "we request that recording participants incorporate characteristics"

(6) The x-axes in figures 4. and 5. use a different order of text-level errors. It makes it hard to interpret those plots. Furthermore, they are called correlation distributions, but I think they are just 2-d histograms with counts of errors. The text should be updated to reflect this.

**Pros:**
   - An exciting new way to evaluate and compare ASR systems
   - A new data resource to enable ASR model comparison using different noise types (background and speech)
   - Generally well-written save for the items mentioned in the previous section
   - The noise portion of the dataset is easily applicable in the analysis of new models, as demonstrated by the rapid feedback from the authors.

**Cons:**
   - Almost none
   - The proposed method currently relies on human annotation of text errors as far as I can tell which limit applicability of the text errors to other models.

---

### Decision · Program_Chairs · 2023-10-07

**Decision:**

Accept-Main

**Comment:**

**Originality**

This paper is very original and proposes evaluating ASR systems in fairly novel way. It enables this evaluation by presenting a dataset with fine-grained annotations for noise types (both background and "speaker" noise).

**Significance**

The topic of explainability in ASR, which this paper is among the first to address, is very important. The resource that the authors have created can easily be used in conjunction with deletion, insertion and substitution errors to help diagnose ASR errors in the presence of specific source noises or speech characteristics. However, it does not appear that automatic classification of text errors using ChatGPT is feasible, so there may be limits the extent to which this error analysis can scale to other ASR systems.

**Clarity**

The paper is generally well structured, and most details are explained. Most reviewers agreed the paper was well-written, however, as the meta-reviewer, I did want to inform the authors of a number of grammatical, spelling or other errors that I found while reading the paper, and which I found made it slightly difficult follow at times, but which are easily fixable.

(1). "Our motivation for proposing a KEBAP, which contains both aspects, is detailed below." -- Which two aspects? It was not clear.

(2) "Since benchmarks measure performance with quantitative metrics, it is crucial **to fine-grain** characteristics for a more detailed diagnosis" -- Perhaps there was a missing word here? it is crucial to **examine**?

(3) "Hence, to solve the explainability issue, we must define **the** error type criteria that consider both the speech- and text-level and create benchmarks to achieve human-level explainability."  -- remove **the**

(4) GEC acronym in 2.3 is not defined until the subsequent section

(4) "we selectively compose **a** text-level error types dataset by human evaluation."

Step 1: Build Text-Level Error Corpus (An example would be nice for clarification)

(5) "we request the recording participants to incorporate characteristics" --> "we request that recording participants incorporate characteristics"

(6) The x-axes in figures 4. and 5. use a different order of text-level errors. It makes it hard to interpret those plots. Furthermore, they are called correlation distributions, but I think they are just 2-d histograms with counts of errors. The text should be updated to reflect this.

**Pros:**
   - An exciting new way to evaluate and compare ASR systems
   - A new data resource to enable ASR model comparison using different noise types (background and speech)
   - Generally well-written save for the items mentioned in the previous section
   - The noise portion of the dataset is easily applicable in the analysis of new models, as demonstrated by the rapid feedback from the authors.

**Cons:**
   - Almost none
   - The proposed method currently relies on human annotation of text errors as far as I can tell which limit applicability of the text errors to other models.